# The role of spin in the degradation of organic photovoltaics

Ivan Ramirez[1 ✉], Alberto Privitera [2], Safakath Karuthedath [3], Anna Jungbluth[2], Johannes Benduhn [4], Andreas Sperlich [5], Donato Spoltore[4], Koen Vandewal [6], Frédéric Laquai [3] & Moritz Riede [2 ✉]

Stability is now a critical factor in the commercialization of organic photovoltaic (OPV) devices. Both extrinsic stability to oxygen and water and intrinsic stability to light and heat in inert conditions must be achieved. Triplet states are known to be problematic in both cases, leading to singlet oxygen production or fullerene dimerization. The latter is thought to proceed from unquenched singlet excitons that have undergone intersystem crossing (ISC). Instead, we show that in bulk heterojunction (BHJ) solar cells the photo-degradation of $C_{60}$ via photo-oligomerization occurs primarily via back-hole transfer (BHT) from a charge-transfer state to a $C_{60}$ excited triplet state. We demonstrate this to be the principal pathway from a combination of steady-state optoelectronic measurements, time-resolved electron paramagnetic resonance, and temperature-dependent transient absorption spectroscopy on model systems. BHT is a much more serious concern than ISC because it cannot be mitigated by improved exciton quenching, obtained for example by a finer BHJ morphology. As BHT is not specific to fullerenes, our results suggest that the role of electron and hole back transfer in the degradation of BHJs should also be carefully considered when designing stable OPV devices.

[1] Heliatek GmbH, Treidlerstrasse 3, 01139 Dresden, Germany. [2] Clarendon Laboratory, Department of Physics, University of Oxford, Parks Road, OX1 3PU Oxford, UK. [3] KAUST Solar Center (KSC), Physical Sciences and Engineering Division (PSE), King Abdullah University of Science and Technology (KAUST), 23955-6900 Thuwai, Saudi Arabia. [4] Dresden Integrated Center for Applied Physics and Photonic Materials (IAPP) and Institute for Applied Physics, Technische Universität Dresden, Nöthnitzer Strasse 61, 01187 Dresden, Germany. [5] Experimental Physics 6, Julius Maximilian University of Würzburg, Am Hubland, 97074 Würzburg, Germany. [6] Institute for Materials Research (IMO-IMOMEC), Hasselt University, Wetenschapspark 1, 3590 Diepenbeek, Belgium. ✉email: ivan.ramirez@heliatek.com; moritz.riede@physics.ox.ac.uk

Given the high power-conversion efficiencies above 18% now reported for organic photovoltaic (OPV) devices, improvements in scalability and long-term device stability are acquiring a new urgency[1–3]. The origins of the degradation in device performance with time have been linked to atmospheric, thermal, or illumination stress[3–5]. Particularly damaging are intrinsic thermal or photo-instabilities in the bulk heterojunction (BHJ) itself, which cannot be mitigated by better encapsulation barriers. One such example is the photo-dimerization and photo-oligomerization of fullerenes, which has been reported to cause substantial losses in performance for a number of polymer: $PC_{61}BM$ BHJs and small molecule/$C_{60}$ planar heterojunctions[6–9]. The photo-oligomerization of $C_{60}$ (henceforth "dimerization"), first reported by Ecklund and colleagues[10], is well documented. Upon irradiation of neat $C_{60}$ films, Raman and mass spectroscopy have shown the formation of $C_{60}$ oligomers with up to 20 repeat units. Based on the short $C_{60}$ singlet lifetime, the observation that the reaction is quenched in oxygen, and the linear increase in the dimerization rate with illumination intensity, it was suggested that dimerization proceeds via a $[2 + 2]$ cycloaddition mediated by the lowest energy $C_{60}$ triplet exciton $T_1$[10–14]. The reaction is understood to proceed identically for $PC_{61}BM$ and $C_{70}$, although their steric groups and lower symmetry impede the bond-face alignment required for cycloaddition[15,16]. These topo-chemical considerations explain the smaller chain lengths obtained for these fullerenes as well as the reaction's suppression at lower temperatures[12]. Heumüller et al.[17] and Pont et al.[18] have investigated the role of polymers in determining the dimerization rate and fraction of $PC_{61}BM$ undergoing dimerization in BHJs. They concluded that $PC_{61}BM$ crystallinity and $PC_{61}BM$ exciton quenching time dictate the reaction kinetics. From studies on blends with polystyrene, an opto-electronically inactive polymer, the fullerene domain size was also found to dictate the fraction of $PC_{61}BM$ which reacts[18]. Morphology thus plays a crucial role and must be carefully accounted for in any dimerization mechanism study. Interestingly, it was also observed that the reaction rate depends on device bias voltage[17]. The loss in device performance was found to be highest for devices aged at $V_{oc}$ but still significant for those kept at $J_{sc}$. This voltage dependence is not a priori consistent with the current hypothesis that unquenched excitons cause the photo-transformation (likely after undergoing intersystem crossing, ISC)[17,18].

In this study, we consider back-hole transfer (BHT) from a charge transfer (CT) state as an alternative pathway for dimerization. When the lowest local triplet excited state lies lower in energy than the CT state, back electron transfer (BET) or BHT from the CT state to form a triplet state on the donor or acceptor is energetically favorable[19]. As re-dissociation of this triplet is not normally energetically possible, BET and BHT typically result in recombination losses. BHT has been significantly less scrutinized than BET but is gaining interest in the context of non-fullerene acceptors (NFAs)[20–22]. Supplementary Fig. 1 illustrates the various scenarios in which BET or BHT dominates. The extent of BET-induced losses remains debated: time-resolved investigations have found BET can represent a significant loss that is difficult to avoid[19,23–26], whereas steady-state measurements suggest it does not affect the non-radiative open-circuit voltage ($V_{oc}$) loss[27]. As triplets are a precursor to highly reactive singlet oxygen, BET has also received attention in the context of atmospheric degradation[28–30]. However, it has not been considered in the context of the more problematic intrinsic degradation processes.

To provide a more comprehensive study of the dimerization mechanism and assess whether rapid exciton quenching is indeed sufficient to avoid dimerization or whether BHT plays a role, we here use model dilute BHJs with a donor (D):$C_{60}$ ratio of only 6% molar. As the blend morphologies at such high fullerene contents

essentially consist of isolated donors surrounded by $C_{60}$ clusters and are remarkably insensitive to the donor choice, this allows us to account for the morphology and to separate it from the influence of donor energetics[31–35]. Working with these model OPV systems, we combine steady-state and time-resolved measurements to show that $C_{60}$ triplets are responsible for the photo-degradation. Crucially we find that the reaction can go ahead despite efficient exciton quenching and negligible ISC yields. We demonstrate using time-resolved electron paramagnetic resonance (TREPR) and temperature-dependent transient absorption (TA) spectroscopy that for systems where the CT energy $E_{CT}$ is larger than the $T_1$ energy $E_{T1}$, dimerization occurs through BHT. A degradation process resulting from BHT cannot be avoided by simply tuning the morphology or improving exciton quenching, making dimerization a much more serious concern than previously thought. As this triplet formation pathway is not specific to fullerenes and $[2 + 2]$ cycloadditions are thought to occur in a wide range of materials[36], BHT or BET could well mediate other intrinsic degradation processes in a variety of systems, including those using high-performance NFAs.

## Results

**Dimerization and CT energy.** Upon dimerization, $C_{60}$ and $PC_{61}BM$ films exhibit a new ultraviolet visible (UV-vis) absorption feature around 320 nm, which has been directly linked to the chemical transformation from high-performance liquid chromatography and Raman spectroscopy[6,7,37]. The tracking of this signature represents the most straightforward and reliable method to study dimerization and directly correlates with the dimer fraction[18]. Figure 1a, b show the evolution of the absorbance of dilute (6% molar) TAPC:$C_{60}$ and m-MTDATA:$C_{60}$ thin films during exposure to white LED light under inert conditions (TAPC: 1,1-bis[4-(N,N-di-ptolylamino)phenyl]cyclohexane, m-MTDATA:     4,4',4'-'Tris(3-m-tolyl-phenylamino)triphenylamine). Material structures and names are provided in Supplementary Table 1. The 6% molar ratio is chosen to obtain direct comparability with the commonly reported 5% wt. TAPC:$C_{60}$ blend.

Neither TAPC nor m-MTDATA have significant absorption in the visible, ensuring that only $C_{60}$ is photo-excited. With TAPC as donor, a change in absorption at 320 nm, characteristic of dimerization, is clearly observed within the first hour of illumination. A fast change is similarly found for BHJs with 37% TAPC (Supplementary Fig. 3). By contrast, there is practically no absorption change at 320 nm after 128 h with m-MTDATA as donor. The evolution of the 320 nm feature with absorbed photon flux and time for these two blends is compared to that of neat $C_{60}$ in Fig. 1c. The photo-transformation is well-described by an exponential, with an effective time constant of 3.6 h for neat $C_{60}$, 8 h for TAPC:$C_{60}$, and 4.1 h for m-MTDATA:$C_{60}$ dilute blend. As the number of dimers depends on the number of photons absorbed, the effective times are scaled to account for the different number of photons absorbed per unit time in each film (see Methods).

The magnitude of the change in absorption at 320 nm is comparable for neat $C_{60}$ and TAPC:$C_{60}$ but much smaller for m-MTDATA:$C_{60}$. The fraction of fullerenes that dimerize is therefore very similar for neat $C_{60}$ and TAPC:$C_{60}$, and much lower for m-MTDATA:$C_{60}$[18]. As it has been shown a number of times that the inclusion of small amounts of donor in $C_{60}$ leads to comparable morphologies[31–34], the difference between the blend photo-stabilities cannot be explained by a morphological effect. The very high internal quantum efficiencies (IQEs) of 85%–90% achieved by TAPC:$C_{60}$ (Supplementary Fig. 14) dilute blends imply a very fast $S_1$ exciton quenching via hole transfer (HT) to a

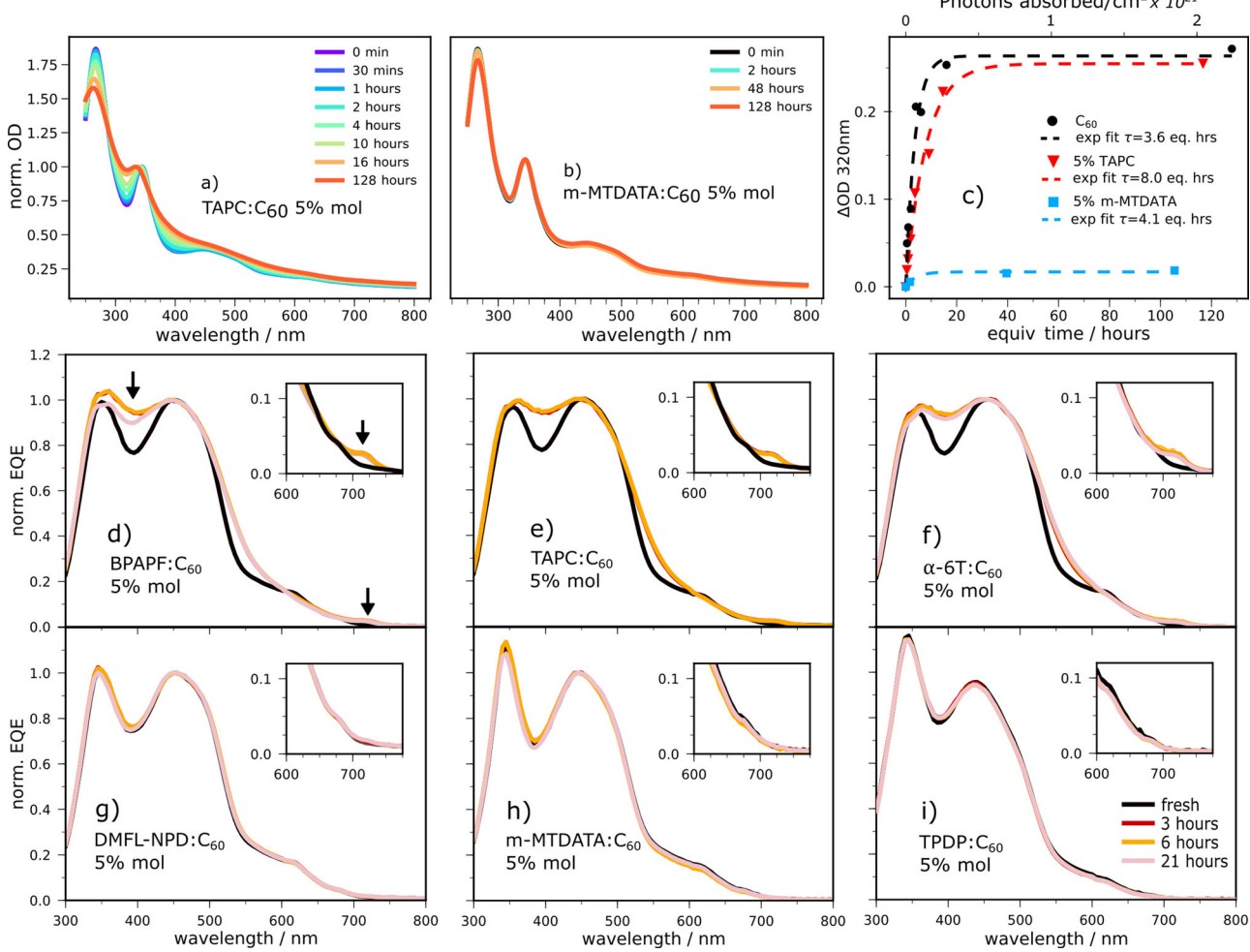

**Fig. 1 Effect of light exposure on blend films and full devices. a**, **b** UV-vis absorption of dilute TAPC:$C_{60}$ and dilute m-MTDATA:$C_{60}$ solar cells exposed to white LED light under inert conditions. **c** Corresponding change in optical density (OD) at 320 nm associated with $C_{60}$ dimerization (neat $C_{60}$ UV-vis time series in Supplementary Methods Fig. 2). The time axis has been scaled to account for the different absorption of each film (see Methods, neat $C_{60}$ 1 h = 1 eq. h). **d**–**i** Change in EQE (normalized to the peak near 450 nm) of full devices with exposure to simulated sunlight (~1.5 suns intensity). The arrows in **d** indicate the 395 and 715 nm spectral features discussed in the text. Insets: detail around 715 nm. Data for $F_4ZnPc:C_{60}$ can be found in Supplementary Fig. 4.

CT state. As $C_{60}$ is the sole absorber in TAPC:$C_{60}$ dilute blends, electron transfer does not play a role. Moreover, a 90% IQE requires a 90% or greater reduction in unquenched $C_{60}$ excitons. Therefore, the comparable dimer yields in TAPC:$C_{60}$ and neat $C_{60}$ are not compatible with unquenched excitons being responsible for dimerization. As detailed in Supplementary Note 6, the increase in fit time constant from 3.6 to 8 h is further evidence that unquenched excitons do not play a significant role in the dimerization of TAPC:$C_{60}$ dilute blends and suggests a different pathway. By contrast, the minor change in absorption observed with m-MTDATA:$C_{60}$ occurs at a rate (4.1 eq. h) nearly identical to the photo-transformation in neat $C_{60}$ (3.6 eq. h) and is consistent with a small remaining fraction of unquenched excitons causing the reaction for this blend.

To better understand the dramatic influence of the donor on photostability, we degrade seven dilute BHJs with different donors and a fixed D:$C_{60}$ molar ratio of 6%. The donors are chosen to cover CT energetics above and below the $C_{60}$ $T_1$, i.e., 1.43–1.5 eV[38,39], as well as a wide range of hole mobilities and CT reorganization energies $\lambda_{CT}$. These properties are summarized in Table 1[40]. To approach real operating conditions, we monitor the spectral change via the external quantum efficiency (EQE) of full

OPV devices exposed to sunlight simulated by a xenon lamp. Although device contacts can degrade under UV or thermal stress[7,41], we show in Supplementary Note 14 that this does not affect our analysis of dimerization in the various blends. The current–voltage (JV) curves of dilute devices before degradation are also included in Supplementary Fig. 13. The peculiarities of these devices and the role of donor have been discussed extensively elsewhere[31,32,40,42–44]. In short, a reduced number of interfaces leads to a reduced density of CT states and increased $V_{oc}$[31], whereas $C_{60}$ delocalization ensures efficient charge dissociation. The precise dissociation efficiency depends strongly on the CT state lifetime, which is primarily dictated by $E_{CT}-\lambda_{CT}$[40,42]. Hole transport is generally argued to occur via tunneling but re-injection to the fullerene has also been suggested[32,33,43,45,46]. For our purposes, these devices provide a convenient model system in which the donor influence can be varied without significantly altering the morphology.

The evolution of EQE spectra with light exposure time is presented in Fig. 1d–i for the various D:$C_{60}$ dilute systems ($F_4ZnPc:C_{60}$, a special case, is shown in Supplementary Fig. 4 and discussed below). For TAPC:$C_{60}$, a rapid change in absorption is again observed. In full devices, the 320 nm feature cannot be

**Table 1 Summary of the dilute blend properties obtained from sEQE and SCLC, and TREPR results.**

| Donor | $E_{CT}$ /eV | $E_{CT}-E_{T1}$/eV | $\lambda_{CT}^{exp}$ /meV | $\mu_h^{dilute}$ /cm$^2$ V$^{-1}$ s$^{-1}$ | Dimerization | TREPR @80 K |
|---|---|---|---|---|---|---|
| TPDP | 0.91[40] | −0.517 | 164[40] | $1.6 \pm 0.5 \times 10^{-5}$[43] | No | - |
| m-MTDATA | 0.95 | −0.470 | 401 ± 15 | $1.1 \pm 0.3 \times 10^{-8}$[43] | No | No signal |
| DMFL-NPD | 1.27 | −0.150 | 240 ± 5 | $2.6 \pm 0.8 \times 10^{-7}$ | No | CT |
| C$_{60}$ (neat) | | $E_{T1} = 1.43$-1.5 eV[38, 39] | | $1.1 \pm 0.3 \times 10^{-5}$ | Yes | ISC, BHT |
| TAPC | 1.45 | 0.020 | 167 ± 3 | $4.8 \pm 1.5 \times 10^{-5}$ | Yes | CT, ISC, BHT |
| α-6T | 1.5 | 0.070 | 279 ± 8 | $2.5 \pm 0.8 \times 10^{-6}$ | Yes | CT, ISC, BHT |
| BPAPF | 1.55 | 0.120 | 180 ± 10 | $3.8 \pm 10^{-5}$ | Yes | CT, ISC, BHT |
| F$_4$ZnPc | 1.54[53] | 0.110 | - | $2.2 \pm 0.7 \times 10^{-6}$ | Yes | BET |

For DMFL-NPD, the TREPR CT signal is weak and only observed when a 410 nm pump is used instead of 532 nm (100-fold higher signal). No signal is obtained at either wavelength for m-MTDATA. $E_{T1}$ is the energy of the lowest excited C$_{60}$ triplet state. The BHT in neat C$_{60}$ is from charges generated from spontaneous dissociation of intermolecular excitons with CT character[14]. Errors in $E_{CT}$ fits are below 3 meV.

detected due to the glass absorption, however, from Fig. 1a), the absorbance at 320 and 395 nm is found to evolve at similar rates for high C$_{60}$ concentrations (Supplementary Fig. 5). We therefore take the change at 395 nm to be directly indicative of dimerization (marked with an arrow in Fig. 1d). In addition, the increased sensitivity afforded by EQE measurements reveals a new sub-gap feature around 715 nm after light exposure, which is not visible in UV-vis spectra (marked with an arrow in Fig. 1d). This change coincides with the C$_{60}$ S$_0$–S$_1$ transition, which is symmetry-forbidden in pristine C$_{60}$[47,48]. We therefore assign the absorption increase at 715 nm to a decrease in wavefunction symmetry upon C$_{60}$ dimerization resulting in a new or more allowed transition, in accordance with previous theoretical calculations[49]. As with TAPC:C$_{60}$, EQE increases at 395 and 715 nm are observed after short exposures to sunlight for the dilute BPAPF (9,9-bis[4-(N,N-bis-biphenyl-4-yl-amino)phenyl]-9H-fluorene), F$_4$ZnPc and α-6T blends. It is noteworthy that unlike the other donors, F$_4$ZnPc is not transparent to sunlight and contributes to the EQE. α-6T contributes only very weakly to the absorption at 6% molar D:C$_{60}$ ratios as its aggregation is disrupted[50]. By contrast, no significant change is observed after 21 h ageing when m-MTDATA, TPDP (2,2',6,'6-tetraphenyl-4,4'-bipyranylidene) or DMFL-NPD (9,9-dimethyl-N,N'-diphenyl-N,N'-di-m-tolyl-9H-fluorene-2,7-diamine) are used as donor. Additional sensitive EQE measurements (Supplementary Fig. 6) before and after 100 h of aging confirm these results. The extra sensitivity also reveals the appearance of the 715 nm feature in neat C$_{60}$ devices after ageing, which is only minor for m-MTDATA:C$_{60}$. The CT energy does not change upon dimerization, though a C$_{60}$ gap narrowing occurs[7].

Table 1 summarizes the properties of the studied dilute blends extracted from sensitive EQE and space charge limited current (SCLC) measurements[51]. As the dilute architecture ensures only minor variations in electron mobility[42,52], C$_{60}$ domain size, and C$_{60}$ crystallinity[35], and because the donor mobility does not correlate with the presence of dimerization (Table 1), both morphology and charge carrier mobilities can be excluded as factors in the photo-degradation. We also exclude any significant morphological changes during degradation as the CT energy is constant and the dimerization induced lattice contraction small (Supplementary Note 3)[10]. By contrast, a clear trend is identified with respect to the relative energetic position of the CT state and the C$_{60}$ T$_1$[38,39]. As evidenced in Fig. 2, no significant dimerization is observed for $E_{CT} < E_{T1}$, suggesting that the primary dimerization pathway in the dilute cells is recombination of a CT state to the C$_{60}$ T$_1$. Why $\lambda_{CT}$ does not play a role in the energetic cutoff, despite BHT being an electron transfer process is discussed in Supplementary Note 5.

**Triplet formation pathway.** Given the experimentally observed relation between the $E_{CT}-E_{T1}$ offset and dimerization rate, we

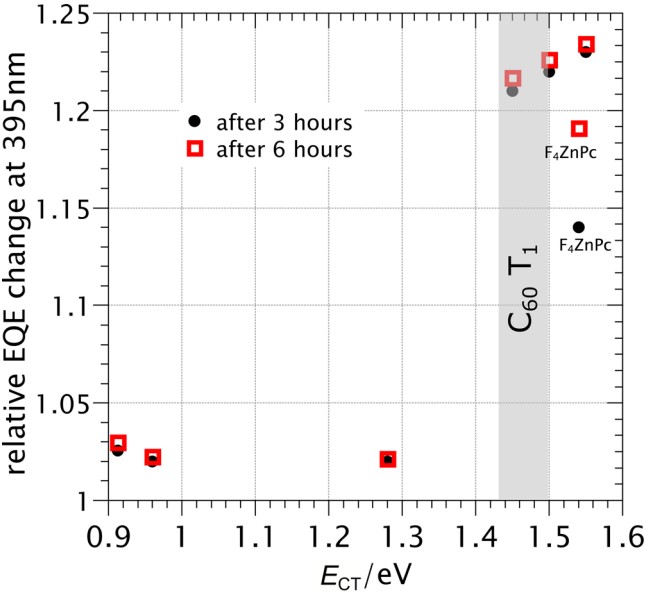

**Fig. 2 Influence of CT energy on dimerization.** Relative change in EQE after exposure to simulated sunlight (xenon lamp) for dilute 6% molar D:C$_{60}$ blends with various CT energies. The relative EQE change is defined as the normalized EQE value of Fig. 1 at 395 nm after 3 (black circles) or 6 hours (open red squared) divided by the initial normalized EQE value at 395 nm. The shaded area represents the reported C$_{60}$ T$_1$ energy range[38, 39]. F$_4$ZnPc: C$_{60}$ (annotated) represents a special case, as the donor triplet energy (1.13 eV) is lower than the $E_{T1}$, which is not the case for the other materials[27].

further investigate the presence of triplets in the various blends and their origin, i.e. whether they originate from unquenched (via ISC) or quenched (via BHT) excitons. TREPR spectroscopy is well suited to this because of its high T$_1$ sensitivity and its ability to unambiguously distinguish between triplet formation pathways[54]. This ability stems from the difference in the non-equilibrium populations (spin polarization) of the triplet sublevels ($m_s = \pm 1$, 0) for each pathway leading to markedly different signals[54]. In a TREPR experiment, the EPR signal is recorded as a function of time after a laser pulse, here at 532 nm. To circumvent the limited time resolution of a few hundred nanoseconds, TREPR measurements of BHJs are usually carried out at low temperatures (here 80 K) where kinetics are slower. We select four dilute blends for further investigation: TAPC:C$_{60}$ and α-6T: C$_{60}$ as model systems with positive $E_{CT}-E_{T1}$ offsets and m-MTDATA:C$_{60}$ and DMFL-NPD:C$_{60}$ as model systems with negative $E_{CT}-E_{T1}$ offsets.

Figure 3a, b illustrate the possible triplet formation pathways which TREPR can help distinguish for the two studied cases.

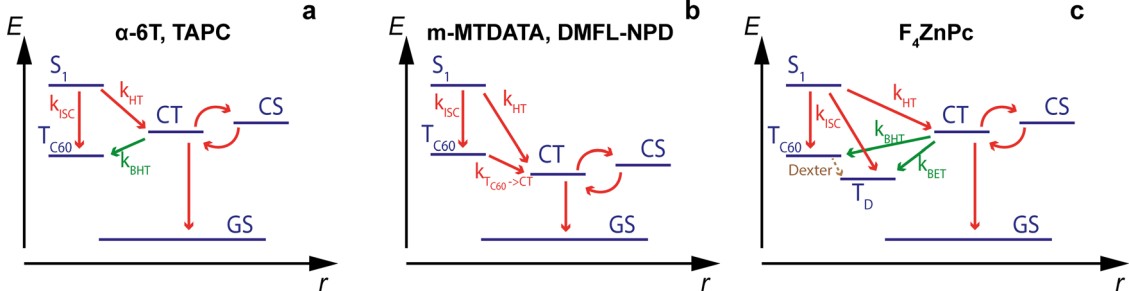

**Fig. 3 Schematic representation of the different photophysical pathways occurring in the studied systems. a** Pathways for $E_{CT} > E_{T1}$, **b** for $E_{CT} < E_{T1}$, and **c** when both the donor and $C_{60}$ Triplets ($T_{C60} = T_1$) are lower in energy than the CT state. The donors for which each case occurs are indicated above and the key rates ($k$) of pathways discussed in the text are labeled.

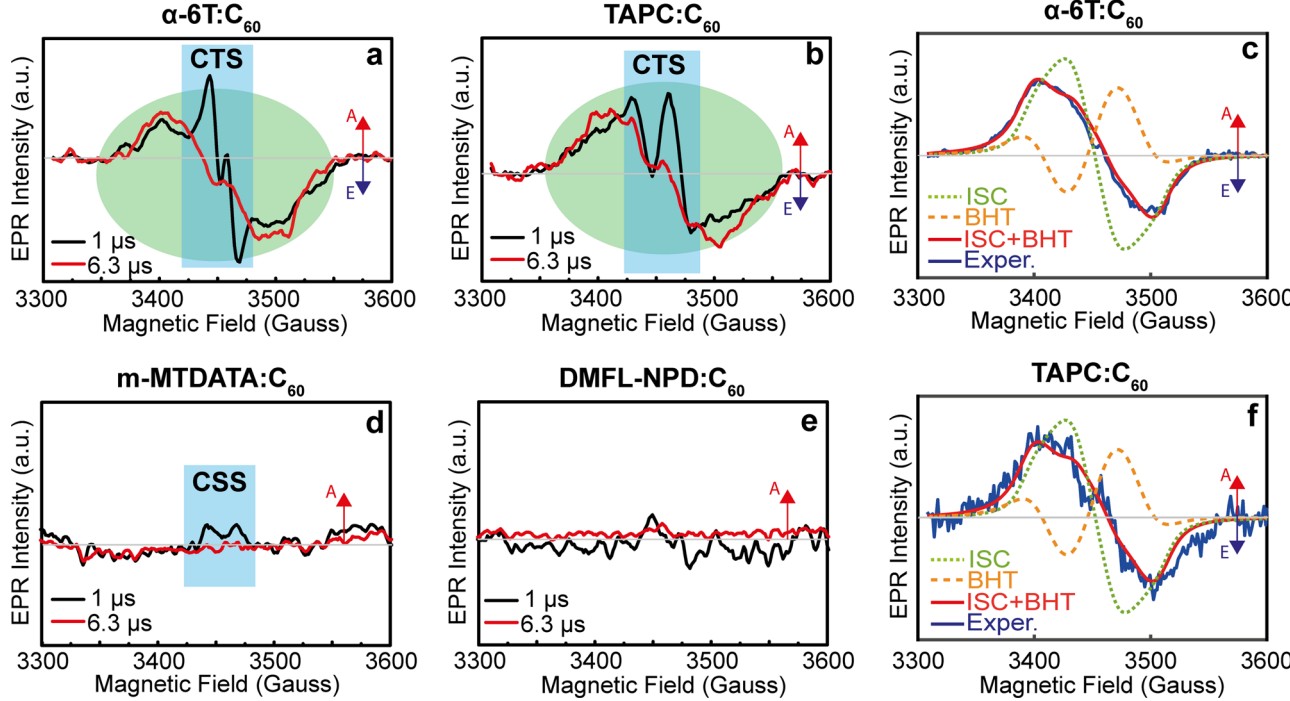

**Fig. 4 TREPR spectra and triplet formation pathways in the dilute (6% molar) blends. a, b, d, e** Smoothed TREPR spectra of dilute $\alpha$-6T:$C_{60}$, TAPC:$C_{60}$, m-MTDATA:$C_{60}$, and DMFL-NPD:$C_{60}$ recorded at 1 (black line) and 6.3 µs (red line) after a 532 nm laser pulse. All the measurements are performed at 80 K. **c, f** Best-fit spectral simulations (red line) of TREPR spectra (blue line) of $\alpha$-6T:$C_{60}$ and TAPC:$C_{60}$ dilute blends taken at 6.3 µs after a 532 nm laser pulse. Two different contributions have been considered in the simulation: intersystem crossing (dotted green line) and back-hole transfer (dashed orange line). The signal regions corresponding to CT states (CTS) and charge separated states (CSS) are highlighted in blue and those corresponding to $C_{60}$ triplets in green in **b**, **c**, and **f**.

After $C_{60}$ photo-excitation and thermalization, $S_1$ quenching via HT ($k_{HT}$) competes with ISC of the $S_1$ to the $C_{60}$ triplet $T_1$. This $T_1$ state is understood to be directly responsible for dimerization and thus a more efficient HT is beneficial for photostability[10,17]. After CT, $T_1$ formation is nonetheless possible if a CT state recombines via BHT. This process is analogous to the more commonly discussed BET to a donor triplet[19]. Because of spin conservation, both BHT and BET require that the CT has some triplet character. How CT states acquire triplet character and undergo BHT to low-lying triplet states has been extensively discussed in the literature and is detailed in Supplementary Note 5[54–56]. In short, triplet $^3$CT formation can occur either via non-geminate recombination of spin-uncorrelated charges or through triplet–singlet mixing in geminate CT states[57], caused, e.g., by hyperfine interactions. In Fig. 3b, BHT is not possible and the HT process from $T_1$ to CT is instead energetically favorable. BET is energetically impossible for these four blends since TAPC, m-MTDATA, and DMFL-NPD are wide (optical) gap materials ($\gtrsim$3 eV) with donor triplet energies ($E_{T,Donor}$) much larger than the measured $E_{CT}$. For $\alpha$-6T, the neat film $E_{T,Donor}$ is 1.5 eV, close to $E_{T1}$[58]. However, the $\alpha$-6T energies are significantly destabilized in dilute blends[44,50]. Donor ISC is not significant in the dilute architecture, because electron transfer is extremely efficient.

In Fig. 4a, b, we report the TREPR spectra of $\alpha$-6T:$C_{60}$ and TAPC:$C_{60}$ dilute blends acquired at 1 and 6.3 µs after a 532 nm laser pulse. The spectra show the presence of two different species. Given their different time evolution it is possible to distinguish between them by considering two different delays after the laser flash. At 1 µs delay, the spectra are mainly characterized by a narrow band signal extending for 70 Gauss with a distorted, double absorption (A)/emission (E) (AEAE) pattern. This signal decays rapidly for both blends and has nearly disappeared after 3 µs. The spectral broadness, the distinctive

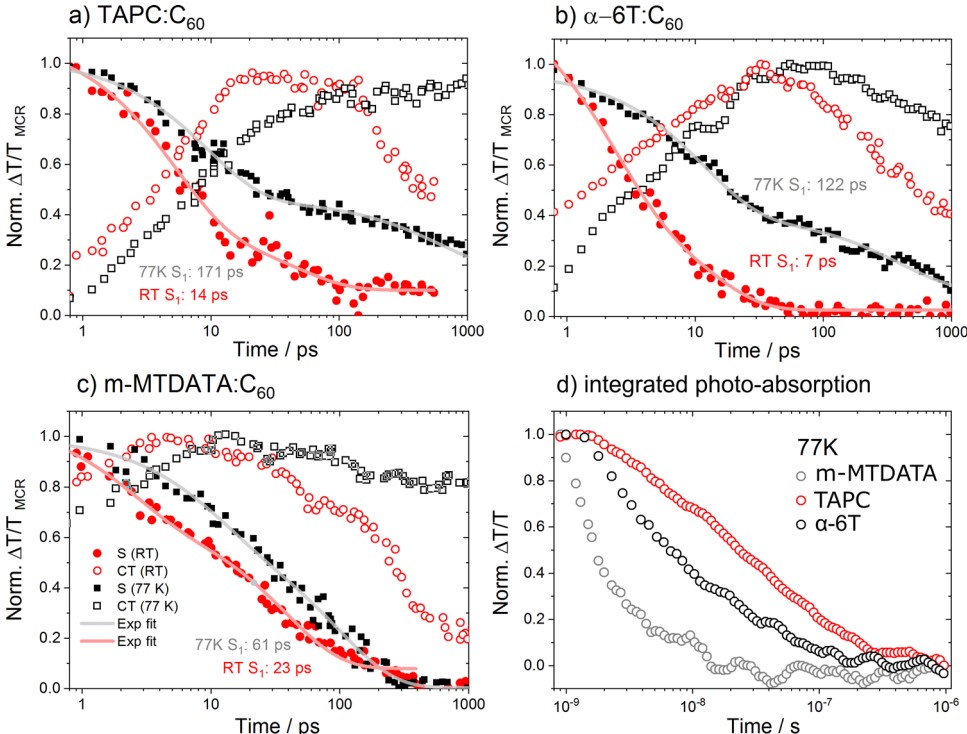

**Fig. 5 Transient absorption kinetics of dilute blends at ambient and cryogenic temperatures. a–c** Picosecond–nanosecond TA kinetics extracted from MCR analysis of TA spectra of dilute blends at room temperature (RT) and 77 K. Closed symbols: singlet state dynamics (labeled S), open symbols: charge + CT dynamics (labeled CT). Black squares: 77 K, red circles: RT. Comparison to raw spectra and signatures extracted from MCR are shown Supplementary Fig. 11. **d** Integrated remaining photo-induced absorption in the 1.8–2 eV region at long time delays (ns–μs) for dilute blends with various donors at 77 K (300 K in Supplementary Fig. 9). The excitation wavelength for both ps-ns and ns–μs TA is 532 nm. The solid lines are bi-exponential fits to the decay (time constants averaged).

AEAE pattern, the field position and the time decay are features commonly observed in TREPR spectra of long-living CT states[59].

At 6.3 μs after the laser pulse, a broader signal, already observed at 1 μs, can be isolated. This signal, extending for ~200 G in enhanced absorption (A) at lower fields and emission (E) at higher fields, can be attributed to the $C_{60}$ triplet exciton[60–62]. To confirm our assignment, we performed best-fit spectral simulations of the two spectra (Fig. 4c, f) using the EasySpin MatLab toolbox[63]. The obtained spectroscopic parameters are reported in Supplementary Table 2 and agree well with the literature values for triplet states localized on $C_{60}$ molecules, confirming our assignment[62,64]. From the raw data and simulations, we observe both ISC and geminate BHT contributions in dilute α-6T:$C_{60}$ and TAPC:$C_{60}$ blends. The ISC polarization is characterized by an AAAEEE pattern typical of $C_{60}$ in solution, where no BHT can occur (dotted green line). BHT polarization follows an AEEAAE pattern (dashed orange line). The overall signal is very well-described by our analysis. TREPR is however not sensitive to BHT/BET originating from non-geminate recombination as for that process there is no spin polarization of the triplet sublevels (Supplementary Note 2). By contrast, TREPR spectra of m-MTDATA:$C_{60}$ and DMFL-NPD:$C_{60}$ (Fig. 4d, e) do not show any appreciable signal either at 1 or at 6.3 μs after the laser flash, although a weak absorptive (A) signal characteristic of separated charges is present in the former (marked CSS, charge separated state).

TA spectroscopy provides an important complement to TREPR thanks to its higher temporal resolution and sensitivity to spin-unpolarized species. TA measurements performed at 77 K with 532 nm excitation (mimicking TREPR conditions) show that recombination occurs significantly faster in m-MTDATA:$C_{60}$ dilute blends than in dilute α-6T:$C_{60}$ or TAPC:$C_{60}$ dilute blends

(Fig. 5d) and much faster than the TREPR temporal resolution. This is in line with the low $E_{CT}-\lambda_{CT}$ of m-MTDATA:$C_{60}$ blends resulting in a fast non-radiative recombination, as per the energy-gap law and recent TA spectroscopy results[40,42]. As dilute DMFL-NPD:$C_{60}$ also has a low $E_{CT}-\lambda_{CT}$, its dynamics are also too fast for TREPR.

**Dynamics at ambient and cryogenic temperatures.** We further use TA spectroscopy at 77 K and 300 K to relate the presence of ISC in our TREPR experiments at 80 K to the degradation process at 300 K. The ISC yield is independent of the CT lifetime (Fig. 3a) and, in our case, depends only on the HT rate, which is typically very fast even at low $S_1$-CT energetic offsets. Significant ISC can only occur if exciton quenching is inefficient. It is therefore not expected in any blend during degradation at 300 K and, as we have argued and TA will further confirm, is inconsistent with our results (Fig. 1c).

To understand why TREPR detects ISC in some blends and not others, we compare the HT dynamics of m-MTDATA, TAPC, and α-6T dilute blends at 300 and 77 K obtained by ultrafast TA spectroscopy. The resulting spectra are shown in Supplementary Fig. 10 and match well with published results[14,42]. The superposition of the excited state spectral signatures makes the TA spectral analysis challenging. To get more insight into the nature of the excited states and their dynamics, we perform multivariate curve resolution–alternating least squares data analysis (MCR-ALS) of the data obtained by TA spectroscopy. MCR-ALS is a soft modeling tool introduced by Tauler and colleagues[26,65–68] and previously used by some of us and others to analyze TA data. Figure 5a–c show the MCR-ALS de-convoluted kinetics of singlets and CT states (including free charges) of α-6T:$C_{60}$,

TAPC:$C_{60}$, and m-MTDATA:$C_{60}$ dilute blends. The spectra of the respective excited species extracted by MCR-ALS are presented in Supplementary Fig. 11. The MCR-ALS signatures obtained for neat $C_{60}$ are also provided and reproduce the previously published signatures well[69–71]. The increase in charge population after 300 ps found for TAPC:$C_{60}$ at 77 K is an artifact originating from the closely matching TAPC cation and $C_{60}$ triplet signatures[14,42,72]. At ambient temperatures the extracted singlet lifetimes in dilute α-6T, TAPC and m-MTDATA blends are 7, 14 and 23 ps, respectively. HT is therefore much faster than ISC (around 1 ns) and inconsistent with the presence/absence of dimerization ($S_1$ quenching is slowest for m-MTDATA:$C_{60}$)[73–75]. The situation is very different at 77 K, where the $S_1$ lifetime increases ($k_{HT}$ decreases) for all blends but significantly more for α-6T (122 ps) and TAPC (171 ps) dilute blends than for m-MTDATA (61 ps). This means higher ISC yields are expected for α-6T and TAPC than m-MTDATA at 77 K. However, the difference in $k_{HT}$ is not sufficiently strong to explain the lack of ISC triplets in m-MTDATA TREPR experiments, from which we conclude that efficient HT from triplets to CTs also occurs when $E_{CT} < E_{T1}$[76] (illustrated in Fig. 3b). Given the large fullerene domains at dilute ratios and given that Förster resonant exciton transfer is inefficient in $C_{60}$, we expect $k_{HT}$ to be similar for triplets and singlets.

The combination of EPR and TA results can be summarized as follows: when $E_{CT} > E_{T1}$, BHT is possible at both room and cryogenic temperatures. ISC is not possible at room temperature because HT is extremely efficient (time constant 13 ps or faster). At 77 K, HT is slowed down (to 122 and 171 ps for α-6T and TAPC respectively), which leaves enough time for some ISC to occur and results in a TREPR signal with BHT and ISC contributions. For $E_{CT} \ll E_{T1}$ (m-MTDATA) BHT is possible neither at room nor cryogenic temperatures. HT is again efficient at room temperature (23 ps) but is less significantly slowed down by the temperature decrease (61 ps at 77 K). This results in a lower ISC yield at 77 K. However, the lack of TREPR ISC signal is primarily due to an efficient $T_1$-to-CT quenching pathway. CT recombination at 77 K is faster than the TREPR time resolution because of the small value of $E_{CT} - \lambda_{CT}$ for the m-MTDATA:$C_{60}$ system. The considerably faster dynamics caused by this low CT gap and the presence of the $T_1$–CT quenching pathway mean that for m-MTDATA:$C_{60}$ CT, mobile charges and $T_1$ have already recombined at the earliest TREPR measurement time, leaving a weak spin-unpolarized signal corresponding to long-lived trapped charges. That this signal is less evident in DMFL-NPD:$C_{60}$ may be due to the system's higher hole mobility resulting in a smaller trapped population. Clearly, TREPR dynamics at 80 K are not necessarily representative of those at room temperature and should be treated with caution. With our knowledge of dynamics in both temperature regimes, however, the trace dimerization observed for m-MTDATA:$C_{60}$ 1 : 19 can be safely assigned to residual unquenched $C_{60}$ excitons undergoing ISC. This is consistent with the near-identical reaction rates found for the m-MTDATA dilute blend and neat $C_{60}$ (Fig. 1c).

**BHT rate and kinetic stabilization**. An interesting case occurs when the donor triplet energy ($E_{T,Donor}$) is below $E_{T1}$ ($C_{60}$) as for dilute F$_4$ZnPc:$C_{60}$ blends ($E_{T,Donor} = 1.13$ eV, Fig. 3c)[27]. For this system, TREPR indicates the formation of donor triplets via BET, although some Dexter transfer from $C_{60}$ triplets may also help populate the donor triplet (Fig. 1c and Supplementary Fig. 7). This is not enough to avoid a fast dimerization and the associated changes in the absorption spectra (Fig. 2). Given the short dimerization time constants, only an extremely efficient BET would lead to the few-% performance drop after 1000 h required

for commercial certification[6,17,77]. Such efficient BET would come at the expense of initial performance and might enable triplet-related donor degradation pathways. It therefore does not constitute a viable mitigation strategy. Indeed, the dilute F$_4$ZnPc:$C_{60}$ device has a lower FF than expected from its $E_{CT}$ and hole mobility (Supplementary Fig. 13)[42], which could be due to efficient BET (other factors such as unfavorable electrostatic potentials cannot be excluded).

It is also worth noting that although we have identified blends for which BHT to a $C_{60}$ triplet occurs, this seems to have only a very limited impact on the FF (Supplementary Table 3). The BHT recombination yield can be estimated from Fig. 1c). Using the known $S_1$ lifetime, estimated incident photon flux and an ISC time constant of 1 ns, we estimate from the neat $C_{60}$ film dimerization rate that only one in *ca.* 10,000 absorbed photons leads to a dimer, meaning only one in approximately a thousand triplets reacts before recombining. From this and the reported CT lifetime in TAPC:$C_{60}$ dilute blends[42], we estimate a $k_{BHT} \approx 4 \times 10^7$ s$^{-1}$ in these blends (Supplementary Note 1). This is consistent with triplets not having a significant effect on the FF and means even a very low BHT yield is sufficient for efficient degradation if a triplet-mediated reaction is possible. In other words, kinetic stability is rather unlikely to be achievable and acceptors (donors) should be inherently stable to inert reactions mediated by a single triplet, such as dimerization. By contrast, we expect that second-order processes such as triplet–triplet annihilation, which can produce enough energy to break bonds, can be kinetically avoided.

## Discussion

In summary, we have used model BHJs with 6% donor:$C_{60}$ molar ratio to show that fullerene dimerization can occur even when photo-generated excitons are efficiently quenched. We find that in these model systems the $C_{60}$ triplets mediating the reaction are formed primarily by BHT from a CT state. Unlike ISC, which in our experiments causes few fullerenes to dimerize, BHT cannot be mitigated through faster exciton quenching, e.g. by tuning the BHJ morphology. The BHT mechanism is revealed by studying model dilute systems but is universal to D–A heterojunctions. This makes dimerization a much more serious and difficult to avoid degradation than previously thought.

The presented BHT mechanism also sheds a new light on previous polymer:PC$_{61}$BM results, although care should always be taken when comparing vacuum and solution-processed devices or different fullerenes. We first stipulate that the correlation found by Heumüller et al. between exciton quenching and dimer yield for PCPDTBT:PC$_{61}$BM is underpinned by an $E_{CT}$ lower than $E_{T1}$ in that particular system, making ISC the only possible dimerization pathway there[17,78]. When BHT is possible, improved quenching will not lead to improved stability, as clearly evidenced by our results. Similarly, the previously observed decrease in PC$_{61}$BM dimer yield in the presence of an extraction bias is easily explained if dimerization results from BHT[17]. As there are fewer recombination events at $J_{sc}$ or max power point than at $V_{oc}$, a lower BHT yield is expected. The associated reduction in dimerization is, however, not sufficient to avoid significant degradation[17]. We also point out that a specific donor cannot easily be claimed to prevent dimerization as different processing conditions can result in different CT energies for the same D:A combination[79]. Thus, BHT-mediated dimerization could occur in a dilute BHJ and not at higher mixing ratios, or even vice versa. Furthermore, back CT is not a fully understood process, which may be subject to subtle morphological effects such as energy gradients near interfaces. In BHJs, ISC may constitute an additional dimerization pathway depending on the morphology.

As BHT only occurs when $E_{T1} < E_{CT}$, we find that dimerization is not possible for fullerene containing systems with $E_{CT} \lesssim 1.4$ eV but is a concern for higher $V_{oc}$ systems. This means the $E_{CT}$ of 1.5 eV predicted to give optimal device performance cannot be achieved without a fast degradation for fullerene acceptors[40]. More worryingly, since back electron and HT are not specific to fullerenes, our finding that triplets produced by BHT can cause rapid photo-degradation even under inert conditions has a wider relevance for OPV and should be carefully studied in NFAs. The recent suggestion that triplet-mediated [2 + 2] cycloadditions, of which fullerene dimerization is an example, occur in a wide range of neat organic semi-conductor films upon light exposure reinforces this message[36]. Nevertheless, the significantly improved early-time stability of several NFAs when compared to PC$_{61}$BM suggests it is possible to find acceptors, which are intrinsically stable to triplet-mediated reactions under inert conditions[80–83].

## Methods

**Sample fabrication, degradation, and dimerization characterization**. Solar cells are fabricated in a commercial evaporation tool (Kurt J. Lesker) and glass–glass encapsulated with a getter to avoid extrinsic degradation. The devices have an active area of 6.44 mm$^2$ as defined by the overlap of the pre-patterned indium-doped tin oxide (ITO, cathode) and aluminum (anode) contacts. MoO$_3$ and BPhen are used as electron and hole blocking layers respectively. Material names, structures, and origin are detailed in the Supplementary Methods.

Device photo-degradation is performed with a xenon lamp in a climate-controlled chamber at 65 °C, 40% relative humidity and $V_{oc}$ (Fischer Scientific) with an intensity of ~2 suns.

*External quantum efficiency*. Monochromatic light modulated by a chopper wheel was shined onto the devices through an optical fiber and the resulting current signal measured with a lock-in amplifier (Signal recovery 7265). Reference spectra were captured with a calibrated Si diode from the Fraunhofer ISE CalLab, Freiburg. Sensitive EQE and *JV* measurements are described in the Supplementary Methods.

Samples for UV-vis measurements consist only of the BHJ and are deposited on quartz (Lesker and Creaphys GmbH). Unencapsulated films are exposed to white LED light in a glovebox. UV-vis measurements are performed in air with a Perkin-Elmer Lambda1050 spectrometer. Samples were discarded after the measurement (one sample per ageing time).

For readability, the absorbed photon flux us translated into an equivalent time using

$$\text{rel abs flux} = \int \phi_{\text{LED}}(\lambda) * \left(1 - 10^{-\text{OD}_{\text{blend}}(\lambda)}\right) d\lambda / \int \phi_{\text{LED}}(\lambda) * \left(1 - 10^{-\text{OD}_{\text{C}_{60}}(\lambda)}\right) d\lambda \tag{1}$$

giving scaling factors of C$_{60}$ = 1, TAPC:C$_{60}$ = 0.912, m-MTDATA = 0.824 eq. h/h exposure. The BHT yield estimation is based on the UV-vis kinetics, modeled as

$$N_{\text{dim}}(t) = N_0 \left(1 - e^{-\frac{t}{\tau}}\right) \tag{2}$$

$N_0$ is the number of C$_{60}$ molecules in the film, estimated from the quartz microbalance used during deposition.

From this, the dimer per absorbed photon yield

$$\Phi_{\text{dim}} = \Phi_{\text{T1}\to\text{dim}}\Phi_{\text{T1}} \tag{3}$$

is estimated, where the triplet to dimer yield (estimated from C$_{60}$ neat film kinetics) is taken to be constant for each material (Supplementary Note 1).

As per ref.[43], mobility was estimated from SCLC measurements of ITO/MoO$_3$/dilute blend/MoO$_3$(3 nm)/Ag (100 nm) devices. Dark *JV*s were fit with

$$J = \frac{9}{8} \varepsilon_0 \varepsilon_r \frac{V^2}{d^3} \mu_0 \exp\left(\gamma\sqrt{V/d}\right) \tag{4}$$

where *d* is the active layer thickness, *V* the bias voltage, and $\varepsilon_0\varepsilon_r$ the active layer electric permittivity ($\varepsilon_r$ = 3.9 ± 0.2). The error is dominated by the uncertainty in *d*.

**TREPR measurements**. All TREPR spectra were recorded on a Bruker Elexsys E680 X-band spectrometer, equipped Oxford Instruments CF935O cryostat and ITC503 controller. The EPR signal was recorded after a short laser pulse (Quanta Ray Nd:YAG, $\lambda$ = 532 nm, pulse length = 9 ns; E/pulse = 3 mJ, 20 Hz repetition rate). The TREPR signal was recorded through a Bruker SpecJet transient recorder. The spectra were acquired with 2 mW microwave power and averaging 400 transient signals at each field position.

EPR samples were prepared as thin films of 50 nm thickness deposited on microscope cover glass, cut to a width of 3 mm with a diamond-tipped glass cutter. The samples are placed in quartz EPR tubes, which are sealed in a nitrogen glovebox, such that all EPR measurements are made without air exposure. TREPR

experiments were performed recording the EPR signal after a short laser pulse produced by a Quanta Ray Nd:YAG pulsed laser ($\lambda$ = 532 nm, pulse length = 9 ns; E/pulse = 3 mJ, 20 Hz repetition rate). The TREPR signal was recorded through a Bruker SpecJet transient recorder. The spectra were acquired with 2 mW microwave power and averaging 400 transient signals at each field position. Fifty-nanometer-thick films were deposited on microscope glass and placed in quartz EPR tubes. All EPR measurements were made without air exposure.

The TREPR spectra simulation were performed by using EasySpin's pepper function[63].

**TA spectroscopy**. TA spectroscopy was carried out using a home-built pump-probe setup. The output of a titanium:sapphire amplifier (Coherent LEGEND DUO, 4.5 mJ, 3 kHz, 100 fs) was split into three beams (2, 1, and 1.5 mJ). Two of them were used to separately pump two optical parametric amplifiers (Light Conversion TOPAS Prime). The photophysical processes in this experiment were initiated by an ultrafast laser pulse generated by TOPAS 1 and were probed by broad white light supercontinua generated in a calcium fluoride window upon exposure to few micro joule of the 1300 nm signal from TOPAS 2. Samples were kept under a dynamic vacuum ($10^{-5}$ mbar) in a cryostat (Optistat CFV, Oxford Instruments). The temperature-dependent study at 300–77 K was performed by cooling the cryostat with liquid nitrogen. The temperature was regulated with a MercuryiTC (Oxford Instruments) temperature controller. Additional details can be found in the Supplementary Methods.

*Multivariate curve resolution–alternating least squares*. MCR-ALS is a soft modeling approach used to factor experimentally measured TA data surfaces into their component spectra and respective concentration profiles applying certain physical constraints such as non-negativity of excited state concentrations or non-positivity of spectra[26,65,68,84]. The MCR analysis and application to TA data has recently been reported and reviewed by Howard et al.[85]. We used constraints for the MCR-ALS analysis such as non-negativity of the excited state concentration and used only two components to get satisfactory description of the experimental TA data.

**Reporting summary**. Further information on research design is available in the Nature Research Reporting Summary linked to this article.

## Data availability
The data that support the findings of this study are available from the corresponding author upon reasonable request.

## Code availability
The codes or algorithms used to analyze the data reported in this study are available from the corresponding authors upon reasonable request.

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

## Acknowledgements

We thank Dr. Olaf Zeika for TPDP synthesis, Dr. Josue Martinez-Hardigree for his insights on morphology, and Professor Natalie Banerji for her valuable advice with TA analysis. A.P. thanks Dr. William Myers and the Centre for Advanced ESR (CAESR) located in the Department of Chemistry of the University of Oxford (supported by EPSRC EP/L011972/1). I.R. thanks TU Dresden technicians for help with sample production and Dr. Frederik Nehm for help with device degradation. This work was supported by European Union's Horizon 2020 research and innovation program under Marie Sklodowska Curie Grant agreement number 722651 (SEPOMO) and by the COST Action MP1307 (StableNextSol). M.R. acknowledges funding from an EU FP7 Marie Curie Career Integration Grant (number PCIG14-GA-2013-630864) and STFC Challenge Led Applied Systems Programe (CLASP, Grant number ST/L006294/1). This publication is based upon work supported by the King Abdullah University of Science and Technology (KAUST) Office of Sponsored Research (OSR) under award number OSR-2018-CARF/CCF-3079.

## Author contributions

I.R., D.S., K.V., and M.R. designed and coordinated the project. I.R. and A.J. carried out the degradation experiments, A.P. the TREPR, S.K. the TA, and J.B. the sEQE measurements. A.S. helped with TREPR analysis. Work at the respective institutes was overseen by D.S., K.V., F.L., and M.R. All authors participated in the analysis and manuscript writing.

## Competing interests

I.R. is employed by Heliatek GmbH, which has an interest in the commercialization of stable OPVs. Other authors declare no competing interests.
