## [Peer Review File · Nature Communications]

REVIEWER COMMENTS

Reviewer #1 (Remarks to the Author):

This is a very well executed study demonstrating that fullerene dimerization can be driven from fullerene triplets states generated by back electron / hole transfer. The results are novel and insightful, with the EPR data showing spin polarised triplets characteristic of charge recombination being particularly novel. Overall this a well executed study suitable for publication in Nature Comm. My only disappointment is that the study does not demonstrate the importance of this degradation pathways in devices with blend compositions typically used for efficient solar cells. For example as BET is far more dominant at open circuit than short circuit, this should mean that intrinsic device degradation is faster at OC than SC. In addition, at MPP, charge extraction is only marginally less efficient than at SC, which should mean this pathway would be strongly suppressed at MPP. The authors should ideally add some data on this point, or at least demonstrate evidence from the literature supporting the relevance of their study to efficient devices working at MPP. I also note that whilst the authors suggest this pathway may be relevant to devices with NFAs, several studies have indicated that devices with some NFAs show better intrinsic stability (and in particular less burn in) than devices with PCBM, indicating suppression of intrinsic degradation pathways such as dimerisation. This should also be commented on.

Reviewer #3 (Remarks to the Author):

1. Summary

In this manuscript, Ramirez et al. explore the intrinsic photostability of fullerene-based organic photovoltaic devices. The degradation pathway of interest is fullerene dimerization, which is typically considered to occur via intersystem crossing (ISC). The authors show that in addition to ISC, back hole transfer (BHT) from the charge transfer state to an excited C60 triplet state contributes to fullerene dimerization. Dilute donor:C60 blends of donor ratio 6% molar were utilized to ensure consistent morphology between donors. Seven donors were chosen to have ECT above or below C60 T1 with varying hole mobilities and λ_{CT} . EQE and UV-vis measurements revealed that there is no significant dimerization when $ECT < ET1$. Expanding upon this, TREPR was performed in cryogenic temperatures on four blends (two with positive and two with negative ECT - ET1 offsets). However, as TREPR is not sensitive to BHT/BET from non-geminate recombination due to lack of spin polarization, TA spectroscopy was utilized as a complementary technique. These experiments revealed that when $ECT > ET1$, BHT and ISC are possible at cryogenic temperatures while only BHT is possible at ambient. On the other hand, when $ECT < ET1$, BHT is not possible at ambient or cryogenic temperatures due to efficient hole transfer. Furthermore, efficient hole transfer from triplets to CTs results in the absence of a TREPR ISC signal, from which the authors concluded that the trace dimerization can be attributed to ISC. Because BHT is possible for fullerene-based systems with $ECT \geq 1.4$ eV, the predicted optimal ECT of 1.5 eV would suffer the associated degradation. Overall, the manuscript is well-written, and presents an alternate degradation pathway that could have ramifications on donor selection for not only fullerene, but also non-fullerene-based systems. However, there are a few issues that should be addressed prior to publication.

2. Comments

- Although polymer:fullerene systems are prevalent in literature, there has been more focus recently on the high PCE of polymer:non-fullerene and the superior mechanical properties of polymer:polymer solar cells. Why is the study limited to only polymer:fullerene systems, especially since BHT is not limited to fullerene-based systems? The authors did mention that "Since this triplet formation pathway is not specific to fullerenes, BHT or BET could well mediate other intrinsic degradation processes in a

variety of systems, including those using high performance non-fullerene acceptors (NFAs)", then why not choosing the NFA to begin with? (in particular, given fullerene is almost obsolete in the field)

- Page 3: "a very fast S1 exciton quenching via hole transfer (HT) to a CT state". Why not via electron transfer to a CT state?
- I would strongly recommend the authors make a clear comparison on BET and BHT, since BHT is a much less discussed process in the literature, and it is in fact the focus of this paper. It would be much appreciated, if the authors could provide a schematic like Scheme 1, in the introduction or the beginning of the results/discussion, to summarize what the community has agreed upon in the past or debated upon (e.g., BET), and what are the photophysical processes are in the systems.
- The study focuses on the dimerization of fullerenes. Why is this degradation pathway chosen specifically?
- Can you elaborate on the potential differences between your vacuum-deposited systems and the more common solution-processed systems? What about the differences between dilute blends and typical BHJs?
- Please explain the choice of 6% molar concentration of donor to represent a dilute blend
- Morphological changes can occur as fullerene dimerization proceeds. Can you confirm that the morphology of the dilute blends remains unchanged during/after testing?
- When degrading full OPV devices, you use simulated sunlight. The UV component can cause degradation of the interlayer (10.1021/acsami.9b04828), which affects EQE. Please address this
- Instead of using a feature at 395 nm which correlates with the desired feature at 320 nm (which is directly indicative of dimerization), would it be possible to use a substrate transparent in the region of interest, such as quartz?
- You mentioned α -6T contributes to absorption and that F4ZnPc is not transparent to sunlight. Were there no other donors that met the desired criteria?
- DMFL-NPD:C60 not discussed or addressed in the TA section
- Please include error bars where applicable, such as for mobilities in Table 1
- There are inconsistencies in British English vs. American English spellings (i.e. "polarization" and "polarisation" are both used, as are "dimerization" and "dimerisation")
- In Scheme 1, please use T1 instead of TC60, or define them to be the same in the text

Response to reviewers' comments for "The role of spin in the degradation of organic photovoltaics" Ramirez *et al.* (NCOMMS-20-24030)

Reviewer 1

- This is a very well executed study demonstrating that fullerene dimerization can be driven from fullerene triplets states generated by back electron / hole transfer. The results are novel and insightful, with the EPR data showing spin polarised triplets characteristic of charge recombination being particularly novel. Overall this a well executed study suitable for publication in Nature Comm.

We are very thankful for this positive feedback.

- My only disappointment is that the study does not demonstrate the importance of this degradation pathways in devices with blend compositions typically used for efficient solar cells. For example as BET is far more dominant at open circuit than short circuit, this should mean that intrinsic device degradation is faster at OC than SC. In addition, at MPP, charge extraction is only marginally less efficient than at SC, which should mean this pathway would be strongly suppressed at MPP. The authors should ideally add some data on this point, or at least demonstrate evidence from the literature supporting the relevance of their study to efficient devices working at MPP.

The role of dimerization in reducing the performance of BHJs with typical blend compositions has been demonstrated a number of times in the literature (see for example refs 6 and 17 of the original manuscript). We therefore focus on unraveling the dimerization(?) mechanism, which remained incompletely understood.

However, we agree that demonstrating the presence of dimerization for blend compositions typically used in efficient solar cells benefits the manuscript and thank the reviewer for suggesting it. We have now included, as part of the supplementary information (SI), UV-VIS absorption spectra before and after exposure to simulated sunlight for a TAPC:C₆₀ blend processed at conditions typical of high efficiency devices, namely 1:2 D:A ratio and a substrate temperature of 90°C.

The reviewer rightly points out that different electrical conditions will lead to different BHT yields and degradation rates. Such a dependence on device bias was first observed by Heumüller *et al.* (reference 17 of the original manuscript), who found that dimerization proceeds more efficiently at V_{oc} than J_{sc} . However, as they demonstrated for polymer:PCBM blends, the effect is still very significant on the 100 hour time scale even at J_{sc} (which represents the best case scenario).

As highlighted in the original manuscript, the electrical dependence follows naturally from the BHT mechanism but cannot be explained in terms of unquenched excitons. However, that the effect is still significant at MPP was not sufficiently clear in the original manuscript and we thank the reviewer for making us aware of this. This point is now stated in the introduction and conclusion of the revised submission (pages 2 and 10), where bias-dependence is discussed.

Revisions to the manuscript (changes in bold + italicized):

Introduction (p.2): Interestingly, it was also observed that the reaction rate depends on device bias voltage.¹⁷ ***The loss in device performance was found to be highest for devices aged at V_{oc} but still significant for those kept at J_{sc} . This voltage dependence*** is not a priori consistent with the current hypothesis that unquenched excitons cause the photo-transformation (likely after undergoing ISC).^{17,18}

Results p.2: Neither TAPC nor m-MTDATA have significant absorption in the visible ensuring that only C₆₀ is photo-excited. With TAPC as donor, a change in absorption at 320 nm, characteristic of dimerization, is clearly observed within the first hour of illumination. ***A fast change is similarly found for BHJs with 37% TAPC (SI figure S2).*** By contrast, there is practically no absorption change at 320 nm after 128 hours with m-MTDATA as donor

Conclusion p11: Since there are fewer recombination events at J_{sc} ***or max power point*** than ***at*** V_{oc} , a lower BHT yield is expected. ***The associated reduction in dimerization is, however, not sufficient to avoid significant degradation.***¹⁷

- I also note that whilst the authors suggest this pathway may be relevant to devices with NFAs, several studies have indicated that devices with some NFAs show better intrinsic stability (and in particular less burn in) than devices with PCBM, indicating suppression of intrinsic degradation pathways such as dimerisation. This should also be commented on.

As the reviewer notes, there are strong reasons to believe that A-D-A fused core acceptors have a significantly better intrinsic stability than PCBM – especially in terms of early time stability (“burn-in”). This point was acknowledged in the conclusion of the original submission (“*Nonetheless, the significantly improved early-time stability of NFAs when compared to PC₆₁BM suggests it is possible to find acceptors which are intrinsically stable to triplet-mediated reactions under inert conditions.*”) but we agree that the choice of citation did not do justice to the fact several solution-processed NFAs have been found to exhibit better stability than PCBM. This has now been remedied by the addition of references 78-80.

Despite the promising data, it is worth noting the very recent suggestion that triplet-mediated cyclo-addition reactions may be near-universally problematic in polymers and small molecules.¹ While the study only considered neat films, this could make our conclusion that dimerization (a triplet-mediated cyclo-addition) is difficult to avoid even in blends all the more relevant. Following reviewer 1 and 3’s comments, we have amended the manuscript to reflect this.

Revisions to the manuscript (changes in bold italics):

Introduction: Since this triplet formation pathway is not specific to fullerenes ***and [2+2] cycloadditions are thought to occur in a wide-range of materials,***³⁶ BHT or BET could well mediate other intrinsic degradation processes in a variety of systems, including those using high performance non-fullerene acceptors (NFAs).

Conclusion: More worryingly, since back electron and hole transfer are not specific to fullerenes, our finding that triplets produced by BHT can cause rapid photo-degradation even under inert conditions has a wider relevance for OPV and should be carefully studied in NFAs. ***The recent suggestion that triplet-mediated [2+2] cycloadditions, of which fullerene dimerization is an example, occur in a wide range of neat organic semi-conductor films upon light exposure reinforces this message.***³⁶ ***Nevertheless,*** the significantly improved early-time stability of ***several*** NFAs when compared to PC₆₁BM suggests it is possible to find acceptors which are intrinsically stable to triplet-mediated reactions under inert conditions.⁷⁷⁻⁸⁰

Reviewer 3

- Summary
In this manuscript, Ramirez et al. explore the intrinsic photostability of fullerene-based organic photovoltaic devices. The degradation pathway of interest is fullerene dimerization, which is typically considered to occur via intersystem crossing (ISC). The authors show that in addition to ISC, back hole transfer (BHT) from the charge transfer state to an excited C60 triplet state contributes to fullerene dimerization. Dilute donor:C60 blends of donor ratio 6%

molar were utilized to ensure consistent morphology between donors. Seven donors were chosen to have ECT above or below C60 T1 with varying hole mobilities and λ_{CT} . EQE and UV-vis measurements revealed that there is no significant dimerization when $ECT < ET1$. Expanding upon this, TREPR was performed in cryogenic temperatures on four blends (two with positive and two with negative ECT - ET1 offsets). However, as TREPR is not sensitive to BHT/BET from non-geminate recombination due to lack of spin polarization, TA spectroscopy was utilized as a complementary technique. These experiments revealed that when $ECT > ET1$, BHT and ISC are possible at cryogenic temperatures while only BHT is possible at ambient. On the other hand, when $ECT < ET1$, BHT is not possible at ambient or cryogenic temperatures due to efficient hole transfer. Furthermore, efficient hole transfer from triplets to CTs results in the absence of a TREPR ISC signal, from which the authors concluded that the trace dimerization can be attributed to ISC. Because BHT is possible for fullerene-based systems with $ECT \geq 1.4$ eV, the predicted optimal ECT of 1.5 eV would suffer the associated degradation. Overall, the manuscript is well-written, and presents an alternate degradation pathway that could have ramifications on donor selection for not only fullerene, but also non-fullerene-based systems. However, there are a few issues that should be addressed prior to publication.

This is a very good summary of our results and we thank the reviewer for taking the time to recapitulate our findings and detail their concerns. We are particularly happy to see they agree with the importance of our work. As detailed below, we believe our revised manuscript and SI address the highlighted issues.

1. Although polymer:fullerene systems are prevalent in literature, there has been more focus recently on the high PCE of polymer:non-fullerene and the superior mechanical properties of polymer:polymer solar cells. Why is the study limited to only polymer:fullerene systems, especially since BHT is not limited to fullerene-based systems? The authors did mention that “Since this triplet formation pathway is not specific to fullerenes, BHT or BET could well mediate other intrinsic degradation processes in a variety of systems, including those using high performance non-fullerene acceptors (NFAs)”, then why not choosing the NFA to begin with? (in particular, given fullerene is almost obsolete in the field)

We thank the reviewer for giving us the opportunity to clarify this.

The small molecule:C₆₀ systems used in this study present a number of key advantages over polymer:NFAs from an experimental and scientific point of view. First and foremost, arriving at our conclusion that BHT is responsible for the intrinsic degradation observed in C₆₀ BHJs requires the careful separation of the influence of morphology and energetics on the photo-physics. This is only possible by use of the dilute architecture (6% molar donor in C₆₀), which is rather unique. We would not have reached such clear conclusions had we used NFAs. Second, NFAs remain so far much less well understood than C₆₀. The use of a well-known system, in the first instance, makes us much more confident we can account for the myriad effects that can come into play during degradation experiments. Related to this, vacuum processing of sublimed-grade materials ensures that impurities and solvent residues do not play a significant role. This would not have been possible with NFAs, where high efficiencies are reached so far, only with solution processable materials. It is also worth highlighting that BHT was not considered a likely issue prior to our findings. It is only because we were able to carefully separate competing effects in a controlled system that a study of the effect of BHT on NFA stability is clearly needed. We note the importance of such a study is increased by the recent finding that many polymers and small molecules undergo [2+2] cycloadditions akin to fullerene dimerization.¹

We agree to a large extent with the reviewer that NFAs likely represent the future of the field, but C₆₀ will remain of high relevance for vacuum deposited organic solar cells in research and commercialisation at least for some time. PCBM is also used in high efficiency ternary NFA systems.² While NFAs are so far limited to solution-processing, vacuum-processing is arguably closest to large scale OPV commercialisation. In particular synthetic complexity and processing are still significant concerns for industrial applications.

2. Page 3: “a very fast S₁ exciton quenching via hole transfer (HT) to a CT state”. Why not via electron transfer to a CT state?

We thank the reviewer for the question. The reason for this is that TAPC is almost transparent to sunlight. Thus, S₁ states are almost exclusively generated on C₆₀. We have added this precision to the revised manuscript. We also note that given the lack of donor domains, ET is expected to be near instantaneous (<<1ps) in the dilute architecture.

Revisions to the manuscript (changes in bold and italics)

The very high internal quantum efficiencies (IQE) of 85%-90% achieved by TAPC:C₆₀ (see SI) dilute blends imply a very fast S₁ exciton quenching via hole transfer (HT) to a CT state. As C₆₀ is the sole absorber in TAPC:C₆₀ dilute blends, ***electron transfer does not play a role. Moreover***, a 90% IQE requires a 90% or greater reduction in unquenched C₆₀ excitons.

3. I would strongly recommend the authors make a clear comparison on BET and BHT, since BHT is a much less discussed process in the literature, and it is in fact the focus of this paper. It would be much appreciated, if the authors could provide a schematic like Scheme 1, in the introduction or the beginning of the results/discussion, to summarize what the community has agreed upon in the past or debated upon (e.g., BET), and what are the photophysical processes are in the systems.

As the reviewer rightly points out, BHT to a triplet has received little attention in the literature. It is therefore very difficult to make a comparative review – in particular, we are not aware of any debates or consensus.

In our opinion, the lack of attention to BHT comes from a focus of back transfer studies on recombination losses in narrow gap polymer:fullerene BHJs. Such systems tend to have a large driving force and a low-lying CT, which makes BHT an uphill process (dimerization only possible via unquenched excitons). For systems with smaller V_{oc} losses or wider donor gaps, BHT becomes possible again (and consequently degradation via BHT). Examples of such systems include eg. those in refs ³ and ⁴ of this response and virtually all published low V_{oc} loss NFAs in which the NFA has the smaller optical gap. A second possible reason for the historical lack of BHT literature is that, as we have found out, triplets of solid-state C₆₀ are particularly difficult to detect in transient absorption spectroscopy experiments.

We expect BHT will receive increased attention in the future and indeed some NFA works already discuss BHT (though not necessarily under this name).⁵⁻⁷ For now, conclusions differ but as the BHJ systems differ, there is no clearly formed debate yet. In the revised manuscript, we highlight that BHT is gaining traction and provide a simple scheme illustrating the scenarios most relevant to OPV in the revised SI under the heading “*Possibility of back electron and back hole transfer*”.

Revisions to the manuscript (changes in bold and italics):

As re-dissociation of this triplet is not normally energetically possible, BET and BHT typically result in recombination losses. ***BHT has been significantly less scrutinised than BET but is gaining interest in***

*the context of non-fullerene acceptors (NFAs).*²⁰⁻²² The extent of BET-induced losses remains debated: time-resolved investigations have found BET can represent a significant loss that is difficult to avoid,^{19,23-26} while steady-state measurements suggest it does not affect the non-radiative open-circuit voltage (V_{oc}) loss.²⁷

4. The study focuses on the dimerization of fullerenes. Why is this degradation pathway chosen specifically?

As outlined in the manuscript's introduction, "intrinsic" degradation pathways occurring in the absence of atmospheric constituents are extremely problematic for long-term OPV module performance. Of these, fullerene dimerization is one of the best documented and most damaging.⁸⁻¹¹ We therefore considered it a natural starting point for in-depth studies of intrinsic degradation pathways, which are still insufficiently scrutinised given their potential severity. In particular it was not clear how such reactions can proceed in blends and therefore to what extent, and how, they can be avoided.

5. Can you elaborate on the potential differences between your vacuum-deposited systems and the more common solution-processed systems? What about the differences between dilute blends and typical BHJs?

We thank the reviewer for the very interesting point they raise. Given the wide variety of materials and processing techniques within both solution and vacuum processing communities and limited amount of data, it is not easy to provide a general answer, but we have outlined some of the differences below.

In our opinion, there are loosely two main likely differences. The first (and most relevant for dimerization) is that domain sizes and domain purities are typically much smaller in vacuum processed blends than in solution processed ones. For example, the average C_{60} coherence length obtained for dilute TAPC: C_{60} films deposited at RT in our vacuum deposition chamber is just over 6 nm¹². This is well within a C_{60} exciton diffusion length and makes ISC nearly impossible. By contrast, clusters larger than 100nm are commonly reported in AFM/SEM images for non-optimised solution processing conditions.¹³ Thus ISC of PCBM excitons is conceivable even in near-optimised systems. The second is that vacuum thermal deposition limits the maximal molecular weight that can be employed. As a consequence donor intramolecular delocalisation is rather limited compared to polymers blends. We expect this leads to different charge dissociation mechanisms in solution and vacuum processed BHJs. It would be very valuable for both communities to see more direct comparisons, especially regarding domain purity and generation/early time mobilities.

The differences between dilute blends and typical (vacuum processed) BHJs with higher donor ratios have received extensive attention in the literature (refs 28, 29 and 36-39 in the original manuscript) and are summarised on page 4 of the original manuscript. However, the key aspect of hole transport, while referenced, could have been better highlighted. We have now explicitly mentioned the two potential transport mechanisms discussed in literature and added some key references. We hope this satisfactorily answers the reviewer's question.

Revisions to the manuscript: (changes in bold and italics):

The peculiarities of these devices and the role of donor have been discussed extensively elsewhere.^{31,32,40,42-44} In short, a reduced number of interfaces leads to a reduced density of CT states and increased V_{oc} ,¹⁴ while C_{60} delocalisation ensures efficient charge dissociation. The precise dissociation efficiency depends strongly on the CT state lifetime, which is primarily dictated by $E_{CT}-\lambda_{CT}$.^{40,42} ***Hole transport is generally argued to occur via tunnelling but re-injection to the fullerene has also been***

suggested. ^{32,33,43,45,46} For our purposes, these devices provide a convenient model system in which the donor influence can be varied without significantly altering the morphology.

6. Please explain the choice of 6% molar concentration of donor to represent a dilute blend

We thank the reviewer for pointing out that this is not obvious. The 6% molar concentration corresponds to 5% weight ratio of TAPC:C₆₀ (optimal concentration for TAPC¹⁴), which is the most widely reported dilute D:A blend. To maintain direct comparability, we set the TAPC concentration at this value. We have now added a note in the sample fabrication methods to explain this apparently strange value.

Changes to the manuscript: Methods: *The 6% molar ratio was chosen to obtain direct comparability with the commonly reported 5% weight TAPC:C₆₀ dilute blend.*

7. Morphological changes can occur as fullerene dimerization proceeds. Can you confirm that the morphology of the dilute blends remains unchanged during/after testing?

In neat C₆₀, the main morphological effect of oligomerisation is a 0.7 % shrinking in the lattice constant.¹⁵ As recently published by Moore et al., the average C₆₀ coherence length obtained from GIWAXS for 6% mol TAPC:C₆₀ is of circa 60 Å (about 5 C₆₀ molecules)¹². Thus the spacing at a C₆₀ grain boundary will at most change by circa 2*(60 Å)*0.7% = 0.8 Å (approx. half a carbon bond). We do not believe this change allows for donor migration, especially for the transparent donors, which are rather bulky and twisted. The invariance of the sensitive EQE CT bands upon dimerisation also suggests that the donor-C₆₀ distance is unaltered and that the morphology is not significantly affected.

We have added these points to the SI under the heading "*Morphological changes upon dimerization*".

Changes to the manuscript:

Because the dilute architecture ensures only minor variations in electron mobility,^{13, 14} C₆₀ domain size and C₆₀ crystallinity,³⁵ and because the donor mobility does not correlate with the presence of dimerization (table I), both morphology and charge carrier mobilities can be excluded as factors in the photo-degradation. ***We also exclude any significant morphological changes during degradation as the CT energy is constant and the dimerization induced lattice contraction small (SI).***

8. When degrading full OPV devices, you use simulated sunlight. The UV component can cause degradation of the interlayer (10.1021/acsami.9b04828), which affects EQE. Please address this

UV-light can indeed lead to additional degradation pathways, like the one mentioned in 10.1021/acsami.9b04828 and other publications^{16,17} and we thank the reviewer for raising this point. The main UV-sensitive interlayer in our case is MoO₃. Its UV instability has been thoroughly documented in Reference 7 of the main text.¹⁸ It is there shown that the MoO₃ film absorption spectrum does not change below 450nm (figure 1 of the publication), outside of the region of interest. Furthermore, UV-VIS absorption measurements and device degradation experiments show that in actual devices MoO₃ is nearly fully stabilised by the ITO substrate (figures 1c, 2f, S1b and S7b of the publication). In particular, MoO₃ degradation was not found to affect the J_{sc} in complete (PHJ)

devices (pre-irradiation of the ITO/MoO₃ did not lead to any change in degradation). We can therefore safely exclude any influence from this interlayer when determining which blends dimerize.

The BPhen hole blocking interlayer at the Al electrode also tends to crystallise on a time scale slightly slower than the dimerization.¹⁹ This is due to its low glass transition temperature.¹⁶ While there is some reduction in the EQE magnitude due to interlayer degradation this does not affect the EQE spectral shape, which we use to track dimerization (note that the J_{sc} reduction is not significant unless the active layer is very rough¹⁹). The behaviour we observe in neat films and devices is very comparable, which further convinces us that our method is robust.

This discussion is now included in the SI under the heading “Effect of UV irradiation and temperature on interlayers”.

Revisions to the manuscript:

To approach real operating conditions, we monitor the spectral change via the external quantum efficiency (EQE) of full OPV devices exposed to sunlight simulated by a xenon lamp. **While device contacts can degrade under UV or thermal stress^{7,40}, we show in the SI that this does not affect our analysis of dimerization in the various blends.** The current-voltage (JV) curves of dilute devices before degradation are **also** included in the SI (figure S11).

9. Instead of using a feature at 395 nm which correlates with the desired feature at 320 nm (which is directly indicative of dimerization), would it be possible to use a substrate transparent in the region of interest, such as quartz?

Unfortunately, it was not possible to obtain quartz with appropriately patterned ITO for our experiments though we agree this would have been preferable. Fortunately, we are able to show that the feature at 395 can reliably be used to infer about changes at 320nm.

10. You mentioned α -6T contributes to absorption and that F₄ZnPc is not transparent to sunlight. Were there no other donors that met the desired criteria?

At 6% molar, the α -6T is not aggregated²⁰ and barely absorbs in the visible (see EQE in figure 1f, which shows no appreciable difference to the TAPC data). We have updated the text to clarify the α -6T absorption at 6% molar.

F₄ZnPc is primarily included to consider the competing effects of BET and BHT on dimerization. It is tricky to satisfy the associated $E_{T,Donor} < E_{CT}$ and $E_{T,Donor} < E_{T,C60}$ requirements without some visible absorption, especially as donor triplet energies are not always known. We would welcome any suggestions of such molecules for future work.

Overall, the inclusion of absorbing donors strengthens the study in our opinion, especially as α -6T and F₄ZnPc are common materials for vacuum processed devices.

Revisions to the manuscript:

Note that unlike the other donors, F₄ZnPc, is not transparent to sunlight **and contributes to the EQE.** **α -6T** contributes only **very** weakly to the absorption at 6% molar D:C₆₀ ratios **as its aggregation is disrupted.**²⁰

11. DMFL-NPD:C60 not discussed or addressed in the TA section

The TA is primarily intended to clarify our TREPR results. We chose to carry out TA experiments on a select subset of systems, which included m-MTDATA:C₆₀. Given the comparability of the two systems – we use the m-MTDATA results to explain the DFML-NPD dynamics in terms of the gap-law.²¹

12. Please include error bars where applicable, such as for mobilities in Table 1

We thank the reviewer for pointing out this omission. We have now carefully re-done all mobility fits, selecting the devices that gave SCLC behaviour over the widest possible voltage range, and added error analysis for sEQE fits and mobilities. The error in the SCLC zero-field mobility is dominated by the uncertainty in the thickness (ca 10%), which results in a ca 30% error is found for all values. There is still no correlation between mobility and dimerization with the updated values. We have updated the graphs in the SI.

13. There are inconsistencies in British English vs. American English spellings (i.e. “polarization” and “polarisation” are both used, as are “dimerization” and “dimerisation”)

Many thanks for flagging this and apologies for the mistake. We have now opted consistently for American spellings so that “dimerization” is spelled as in previous work.

14. In Scheme 1, please use T1 instead of TC60, or define them to be the same in the text

Many thanks for pointing this, we have now defined them to be the same in the caption. We have also spotted a T_1/T_{C60} inconsistency in scheme 1b) which we have corrected in the revised version.

References

1. Yamilova, O. R. *et al.* What is Killing Organic Photovoltaics: Light-Induced Crosslinking as a General Degradation Pathway of Organic Conjugated Molecules. *Adv. Energy Mater.* **10**, (2020).
2. Lin, Y. *et al.* 17% Efficient Organic Solar Cells Based on Liquid Exfoliated WS₂ as a Replacement for PEDOT:PSS. *Adv. Mater.* **31**, (2019).
3. Chen, Y. *et al.* Vacuum-Deposited Small-Molecule Organic Solar Cells with High Power Conversion Efficiencies by Judicious Molecular Design and Device Optimization. *J. Am. Chem. Soc.* **134**, 13616–13623 (2012).
4. Kawashima, K., Tamai, Y., Ohkita, H., Osaka, I. & Takimiya, K. High-efficiency polymer solar cells with small photon energy loss. *Nat. Commun.* **6**, 1–9 (2015).
5. Du, X. *et al.* Delayed Fluorescence Emitter Enables Near 17% Efficiency Ternary Organic Solar Cells with Enhanced Storage Stability and Reduced Recombination Energy Loss. *Adv. Funct. Mater.* **30**, 1–13 (2020).
6. Kotova, M. S. *et al.* On the absence of triplet exciton loss pathways in non-fullerene acceptor based organic solar cells. *Mater. Horizons* **7**, 1641–1649 (2020).
7. Van Landeghem, M., Lenaerts, R., Kesters, J., Maes, W. & Goovaerts, E. Impact of the donor polymer on recombination: Via triplet excitons in a fullerene-free organic solar cell. *Phys. Chem. Chem. Phys.* **21**, 22999–23008 (2019).
8. Pont, S., Durrant, J. R. & Cabral, J. T. Dynamic PCBM:Dimer Population in Solar Cells under Light and Temperature Fluctuations. *Adv. Energy Mater.* **9**, (2019).
9. Heumüller, T. *et al.* Morphological and electrical control of fullerene dimerization determines organic photovoltaic stability. *Energy Environ. Sci.* **9**, 247–256 (2016).
10. Distler, A. *et al.* The effect of PCBM dimerization on the performance of bulk heterojunction solar cells. *Adv. Energy Mater.* **4**, 1–6 (2014).

11. Distler, A. The Role of Fullerenes in the Photo-degradation of Organic Solar Cells. *Thesis* 118 (2015).
12. Moore, G. J. *et al.* Ultrafast Charge Dynamics in Dilute-Donor versus Highly Intermixed TAPC:C60 Organic Solar Cell Blends. *J. Phys. Chem. Lett.* **11**, 5610–5617 (2020).
13. Hoppe, H. *et al.* Nanoscale morphology of conjugated polymer/fullerene-based bulk-heterojunction solar cells. *Adv. Funct. Mater.* **14**, 1005–1011 (2004).
14. Zhang, M., Wang, H., Tian, H., Geng, Y. & Tang, C. W. Bulk heterojunction photovoltaic cells with low donor concentration. *Adv. Mater.* **23**, 4960–4964 (2011).
15. Rao, a. M. *et al.* Photoinduced polymerization of solid C₆₀ films. *Science (80-.)*. **259**, 955–957 (1993).
16. Burlingame, Q. *et al.* Reliability of Small Molecule Organic Photovoltaics with Electron-Filtering Compound Buffer Layers. *Adv. Energy Mater.* **6**, 1–11 (2016).
17. Jin, F. *et al.* Improvement in power conversion efficiency and long-term lifetime of organic photovoltaic cells by using bathophenanthroline/molybdenum oxide as compound cathode buffer layer. *Sol. Energy Mater. Sol. Cells* **117**, 189–193 (2013).
18. Zhang, H. *et al.* Photochemical transformations in fullerene and molybdenum oxide affect the stability of bilayer organic solar cells. *Adv. Energy Mater.* **5**, 1–9 (2015).
19. Song, B., Burlingame, Q. C., Lee, K. & Forrest, S. R. Reliability of mixed-heterojunction organic photovoltaics grown via organic vapor phase deposition. *Adv. Energy Mater.* **5**, 1–6 (2015).
20. Vandewal, K. *et al.* Absorption tails of donor:C₆₀ blends provide insight into thermally activated charge-transfer processes and polaron relaxation. *J. Am. Chem. Soc.* **139**, 1699–1704 (2017).
21. Benduhn, J. *et al.* Intrinsic non-radiative voltage losses in fullerene-based organic solar cells. *Nat. Energy* **2**, 17053 (2017).

REVIEWERS' COMMENTS

Reviewer #1 (Remarks to the Author):

The authors have carefully and thoroughly addressed my concerns. I am happy to recommend acceptance for publication.

Reviewer #3 (Remarks to the Author):

1. Summary

In this manuscript, Ramirez et al. explore the intrinsic photostability of fullerene-based organic photovoltaic devices. The degradation pathway of interest is fullerene dimerization, which is typically considered to occur via intersystem crossing (ISC). The authors show that in addition to ISC, back hole transfer (BHT) from the charge transfer state to an excited C60 triplet state contributes to fullerene dimerization. Dilute donor:C60 blends were utilized to ensure consistent morphology between donors. Seven donors were chosen to have ECT above or below C60 T1 with varying hole mobilities and λ CT. EQE and UV-vis measurements revealed that there is no significant dimerization when $ECT < ET1$. Expanding upon this, TREPR was performed in cryogenic temperatures on four blends (two with positive and two with negative $ECT - ET1$ offsets). However, as TREPR is not sensitive to BHT/BET from non-geminate recombination due to lack of spin polarization, TA spectroscopy was utilized as a complementary technique. These experiments revealed that when $ECT > ET1$, BHT and ISC are possible at cryogenic temperatures while only BHT is possible at ambient. On the other hand, when $ECT < ET1$, BHT is not possible at ambient or cryogenic temperatures due to efficient hole transfer. Furthermore, efficient hole transfer from triplets to CTs results in the absence of a TREPR ISC signal, from which the authors concluded that the trace dimerization can be attributed to ISC. Because BHT is possible for fullerene-based systems with $ECT \geq 1.4$ eV, the predicted optimal ECT of 1.5 eV would suffer the associated degradation. Overall, the manuscript is well-written, and presents an alternate degradation pathway that could have ramifications on donor selection for not only fullerene, but also non-fullerene-based systems. The revised manuscript, SI, and author rebuttals have clarified previous concerns. Given the remaining typographical errors are addressed, this work is suitable for publication in Nature Communications.

2. Comments

- Maybe reinforce the choice of small molecule:fullerene as the model system. For example, as in the rebuttal, explicitly state that the material and processing used eliminates solvent and impurities, that PCBM is still used in ternary systems, and that vacuum-processing improves scalability
- o Regarding commercialization: The rebuttal mentions that vacuum-processing is the closest to a large-scale commercialization process; however, it can also be argued that roll-to-roll processing—as would be possible with solution-processed materials—is more viable. If the scalability of vacuum processing is mentioned, this should be addressed.
- Perhaps move the note that 6 mol% = 5 wt% to earlier in the main text?
- The use of a dilute system minimizes variables; however, real BHJs are not dilute. Is this system truly an appropriate model for degradation in BHJs, or is it only to demonstrate the importance of the studied degradation pathway? Please clarify in the text.
- Please address remaining inconsistencies in British vs. American English (commercialisation, normalised, summarised, delocalisation, favourable, etc). Not sure whether Nature Comm prefers one way or the other.

Final revisions following reviewer 3's comments

Thank you for your comments and chasing down the last ambiguities and spelling inconsistencies.

- Maybe reinforce the choice of small molecule:fullerene as the model system. For example, as in the rebuttal, explicitly state that the material and processing used eliminates solvent and impurities, that PCBM is still used in ternary systems, and that vacuum-processing improves scalability o Regarding commercialization: The rebuttal mentions that vacuum-processing is the closest to a large-scale commercialization process; however, it can also be argued that roll-to-roll processing—as would be possible with solution-processed materials—is more viable. If the scalability of vacuum processing is mentioned, this should be addressed.

Our answer in the previous revision round was somewhat more nuanced and stated that vacuum processing is “arguably closest to commercialisation”. We agree there are also arguments for solution processing of organic solar cells; however, roll-to-roll fabrication is not one of them as it is used in both large-scale solution and vacuum processing of organic solar cells (e.g. Sunew or Armor and Heliatek).

- Perhaps move the note that 6 mol% = 5 wt% to earlier in the main text?
Sure, now moved to first paragraph of results, which does make more sense.

- The use of a dilute system minimizes variables; however, real BHJs are not dilute. Is this system truly an appropriate model for degradation in BHJs, or is it only to demonstrate the importance of the studied degradation pathway? Please clarify in the text.

We hope we understood the question, in particular what is meant by “appropriate model”. BHT is not unique to dilute cells, but dilute cells are not the same as BHJs, eg in terms of energetics, which can affect degradation. We have tried to further clarify the discussion (changes highlighted in yellow on page 10).

- Please address remaining inconsistencies in British vs. American English (commercialisation, normalised, summarised, delocalisation, favourable, etc). Not sure whether Nature Comm prefers one way or the other.

Thanks! We find these nearly impossible to catch, sorry.